# Overcoming Drug Resistance by Taking Advantage of Physical Principles: Pressurized Intraperitoneal Aerosol Chemotherapy (PIPAC)

**DOI:** 10.3390/cancers12010034

**Published:** 2019-12-20

**Authors:** Giorgi Nadiradze, Philipp Horvath, Yaroslav Sautkin, Rami Archid, Frank-Jürgen Weinreich, Alfred Königsrainer, Marc A. Reymond

**Affiliations:** National Center for Pleura and Peritoneum, University of Tübingen, 72076 Tübingen, Germany; giorgi.nadiradze@med.uni-tuebingen.de (G.N.); philipp.horvath@med.uni-tuebingen.de (P.H.); iaroslav.sautkin@med.uni-tuebingen.de (Y.S.); rami.archid@med.uni-tuebingen.de (R.A.); juergen.weinreich@med.uni-tuebingen.de (F.-J.W.); alfred.koenigsrainer@med.uni-tuebingen.de (A.K.)

**Keywords:** eritoneal metastases, intraperitoneal chemotherapy, aerosol, pressure, chemoresistance

## Abstract

Theoretical considerations as well as comprehensive preclinical and clinical data suggest that optimizing physical parameters of intraperitoneal drug delivery might help to circumvent initial or acquired resistance of peritoneal metastasis (PM) to chemotherapy. Pressurized Intraperitoneal Aerosol Chemotherapy (PIPAC) is a novel minimally invasive drug delivery system systematically addressing the current limitations of intraperitoneal chemotherapy. The rationale behind PIPAC is: (1) optimizing homogeneity of drug distribution by applying an aerosol rather than a liquid solution; (2) applying increased intraperitoneal hydrostatic pressure to counteract elevated intratumoral interstitial fluid pressure; (3) limiting blood outflow during drug application; (4) steering environmental parameters (temperature, pH, electrostatic charge etc.) in the peritoneal cavity for best tissue target effect. In addition, PIPAC allows repeated application and objective assessment of tumor response by comparing biopsies between chemotherapy cycles. Although incompletely understood, the reasons that allow PIPAC to overcome established chemoresistance are probably linked to local dose intensification. All pharmacological data published so far show a superior therapeutic ratio (tissue concentration/dose applied) of PIPAC vs. systemic administration, of PIPAC vs. intraperitoneal liquid chemotherapy, of PIPAC vs. Hyperthermic Intraperitoneal Chemotherapy (HIPEC) or PIPAC vs. laparoscopic HIPEC. In the initial introduction phase, PIPAC has been used in patients who were quite ill and had already failed multiple treatment regimes, but it may not be limited to that group of patients in the future. Rapid diffusion of PIPAC in clinical practice worldwide supports its potential to become a game changer in the treatment of chemoresistant isolated PM of various origins.

## 1. Introduction

Every 30 seconds worldwide, a new patient is diagnosed with peritoneal metastasis (PM), the spreading of cancer cells into the peritoneal cavity [1]. Upon diagnosis, current therapy guidelines [2,3,4] recommend palliative systemic chemotherapy or best supportive care. In spite of recent progress in the fields of targeted therapy, immunotherapy, cytoreductive surgery (CRS) and intraperitoneal chemotherapy, PM is still perceived as a fatal disease [5].

Patients with PM have shorter survival than those with parenchymatous metastasis, for example patients with liver metastasis in colorectal cancer [6]. This poor prognosis is multifactorial and includes chemoresistance to cytotoxic drugs, poor tolerance of chemotherapy, steeper patient performance decline and intestinal dysfunction associated with tumor bowel invasion [6].

Why are PM relatively resistant to chemotherapeutic drugs administered intravenously? The initial hypothesis was that molecular mechanisms were responsible for resistance so that newly developed targeted drugs would overcome this resistance. However, the effect of new drugs on the prognosis of PM patients was limited. For example, in colorectal cancer, treatment-related resistance of PM did not disappear with the introduction of irinotecan and oxaliplatin [7]. Incremental differences in outcomes between non-peritoneal and peritoneal cohorts even increased after the introduction of targeted agents [8]. Similarly, in advanced gastric cancer, where doublet or triplet systemic chemotherapy is the standard-of-care, their ability in achieving long-term survival in patients with PM is still “modest at best” [8].

Thus, beyond molecular mechanisms, other factors must play a role in the chemoresistance of PM [9]. One hypothesis is anticancer drugs not penetrating tissue efficiently [10]. In order to exert a therapeutic effect, drugs must reach all the cancer cells at an appropriate cytotoxic concentration and period of time. This might not be the case for current therapies of PM.

### 1.1. The Peritoneum Is Poorly Vascularized

Only 2–5% of the cardiac minute volume reaches the peritoneum, so most chemotherapy (95–98%) administered intravenously is by-passing the peritoneum and causing systemic toxicity. Moreover, the peritoneal microcirculation is characterized by a low capillary density in comparison to other organs [11,12]. In addition, in PM, local vascular distribution varies between vital and necrotic zones, with some areas receiving no blood supply at all [13]. These added factors result in poor tissue uptake of compounds administered into the systemic blood compartment.

### 1.2. Interstitial Fluid Pressure Is Increased in PM

Fluid pressure is elevated in the interstitial tissue of tumors [14]. For example, interstitial fluid pressure values as high as 33 mmHg have been recorded in some sarcomas [15]. Several studies indicate that high interstitial fluid pressure in the tumor is correlated with poor prognosis [16]. Most drugs used for systemic treatment of patients with cancer—high-molecular-weight compounds in particular—are transported from the circulatory system through the interstitial space by convection, that is, they are carried by streaming of a flowing fluid [17]. Thus, increased interstitial fluid pressure within the tumor node causes less uptake of drugs into the tumor. Moreover, in PM, interstitial fluid pressure profiles vary between different tumor zones [13], suggesting that exposition of tumor cells to chemotherapeutic drugs might vary within the metastasis.

During tumor progression the extracellular matrix undergoes so-called mesothelial–mesenchymal transition (MMT) [18]. MMT is characterized by the progressive replacement of normal peritoneal mesothelial cells tissue by fibroblasts, a major cellular component of scar tissue [19]. During MMT, peritoneal mesothelial cells lose their epithelial-like characteristics, including cell–cell junctions, tight junctions, adherence junctions and desmosomes, lose their apicobasal polarity and progressively acquire a mesenchymal phenotype, characterized by actin reorganization, stress fiber formation, migration, and invasion [20]. These dramatic changes in tissue architecture, formerly named “desmoplasia”, are most pronounced at the tumor invasion front. Due to tissue remodeling and increased interstitial fluid pressure (IFP), drug uptake is less effective into hard, fibrotic tissue than into normal tissue. This results in increased resistance to chemotherapy, which is largely independent of the mode of action of the drug applied.

## 2. Pharmacokinetic Aspects of Intraperitoneal Chemotherapy

Under normal conditions, the peritoneal–plasma barrier prevents absorption of large molecules, including most drugs, from the peritoneal cavity into the systemic blood circulation (so-called peritoneal clearance). Thus, intraperitoneal delivery can reach high local drug concentration within the peritoneal cavity, whereas systemic drug concentrations remain low. From a pharmacological perspective, this is advantageous for treating disease limited to the peritoneal cavity. On the other hand, intravenous drug delivery is relatively ineffective for treating intraperitoneal disease, since intraperitoneal drug concentration remains low in spite of high systemic peak concentration [21]. Thus, the peritoneal–plasma barrier has some functional similarities with the blood–brain barrier, which is barely permeable to chemotherapeutic drugs [22].

## 3. Physical Laws Governing Intraperitoneal Chemotherapy

Intuitively, applying a high intraperitoneal drug concentration is expected to ensure therapeutic success. However, a high intraperitoneal drug concentration is not a synonym of therapeutic success since it does not guarantee drug tissue penetration or achieving cytotoxic concentration of the drug within the PM for an optimal period of time [23]. An unfortunate example is the PRODIGE-7 trial, a Phase 3 randomized study aimed at curing PM of colorectal origin by combining cytoreductive surgery with heated intraperitoneal chemotherapy (EudraCT Number: 2006-006175-20). The intraperitoneal dose of oxaliplatin applied was 460 mg/m^2^ dody surface area (BSA), a dose largely superior to a typical intravenous dose. This dose was determined on the basis of a previous dose-escalation trial and determination of systemic maximal tolerated dose (MTD) [24]. Surprisingly, the PRODIGE-7 trial failed to meet its objectives, probably because oxaliplatin administered as HIPEC has very limited tissue penetration and the application time of 30 min was too short [25]. In an experimental study, compared with the amount of oxaliplatin expected in peritoneal perfusates by calculation, only 10–15% of the parent drug could be detected during HIPEC [23]. This (very) limited drug penetration into the tumor nodes represents a major challenge for future intraperitoneal chemotherapy protocols with HIPEC [21].

A further problem is the incomplete exposure of the peritoneum to cytotoxic drugs when they are administered as liquids [26]. Physical laws say that, assuming a constant temperature and in contrast to a gas, a liquid does not expand within a closed volume. In clinical practice, patients are exposed to gravity and, in the supine position, any liquid distributed into the peritoneal cavity will collect in the posterior abdomen. Thus, to improve distribution, the abdomen is usually filled up with a large volume and the patient is requested to change position. However, all these maneuvers have limited efficacy. Figure 1 illustrates that, in parallel hydraulic systems, the totality of a liquid will flow into the channel with the lowest resistance, even in the case of minimal difference between channel resistance. Resistance to flow varies within the abdomen between different locations, depending on the anatomy of the peritoneal cavity, adhesions, distance between inflow and outflow catheters, etc. Thus, all liquid chemotherapy volume will flow along a proffered path of (minimal) resistance from an input to an output catheter. Physically, homogeneous intraperitoneal drug delivery using liquids (for example, during open or laparoscopic HIPEC) is not possible.

Experimental data obtained in animal models confirm limited exposure of the peritoneal surface during conventional peritoneal lavage: when peritoneal dialysis was carried out in rodents with a solution containing methylene blue and bovine serum albumin, autopsy findings showed that large parts of the visceral and parietal peritoneum displayed no stain or very little stain [27]. In particular, the hidden aspects of the caecum and stomach, as well as large portions of the small and large intestines and of the diaphragm, remained unstained. Our early experiments confirmed that distribution of methylene blue within the peritoneal cavity is poor after peritoneal lavage [28]. Intestinal distribution of luminescent mRNA complexes was examined in a rodent model: the median small bowel luminescent surface was only about 10% after intraperitoneal injection of a liquid solution [29]. After laparoscopic HIPEC, mean oxaliplatin tissue concentration was 17 times higher in the parietal than in the visceral peritoneum in a swine model, with extreme reported values differing by a factor of 20 [30].

## 4. Physical Interventions to Increase Drug Uptake

At least three physical methods have been shown to increase and/or homogenize drug uptake into the peritoneum: increasing hydrostatic pressure, using aerosols and, to a lesser degree, generating hyperthermia.

### 4.1. Increasing Pressure

A large set of experimental data supports the benefit of increasing intraperitoneal hydrostatic pressure for increasing tissue drug uptake. Increased intraabdominal pressure is thought to generate a convective flux that drags the macromolecular drugs along from the peritoneal cavity into the subperitoneal tissue. Jacquet et al. first showed a significant enhancement of doxorubicin uptake in the abdominal wall and diaphragm of rats by increasing intraperitoneal pressure to 20–30 mmHg [31]. Esquis et al. confirmed in a rat tumor model [32] that increasing the intraperitoneal pressure resulted in a significantly higher cisplatin penetration in tumor tissue after HIPEC. In an experiment in a healthy swine model, HIPEC hydrostatic pressure was augmented to 18 mmHg by a water column, resulting in significantly enhanced diffusion of oxaliplatin both in the visceral and parietal peritoneum. Notably, high intraabdominal pressure did not increase systemic drug uptake [33]. In a further study, tissue uptake of oxaliplatin during open vs. closed HIPEC was compared at different pressures (normobaric, 18 mmHg, and 30 mmHg). Whereas oxaliplatin tissue concentration increased with pressure application in the open group, this concentration was found to be consistently lower in the closed than in the open group and this was independent of the pressure applied [34]. Clinical application of HIPEC with elevated intraabdominal pressure had so far been limited to palliating debilitating malignant ascites with laparoscopic HIPEC at 10–15 mmHg [35]. Recent clinical data in ovarian cancer show that laparoscopic HIPEC (under pressure of 12–15 mmHg) significantly increased (doubled) the tissue concentration of cisplatin as compared to open HIPEC (with no hydrostatic pressure applied) [36]. Similarly, when intraabdominal pressure was increased by perfusing additional volume during closed HIPEC, drug concentration was doubled in the visceral peritoneum (but not in the parietal peritoneum) [37].

### 4.2. Use of Aerosols

In contrast to liquids, gases have no definite volume and, in contrast to solids, they have no definite shape. In a perfect gas, molecules travel in random paths and collide with one another and the recipient walls. These collisions exert a pressure per unit area and also cause the gases to homogenously occupy the totality of the volume available. An aerosol is a suspension of liquid droplets (or particles) in a gas. Thus, spatial distribution of an aerosol will be less homogeneous than a gas, but more homogeneous than a liquid (Figure 2).

Physical laws governing deposition of medical aerosols are concerned principally with inertial impaction and gravitational sedimentation. The behavior of therapeutic aerosols is determined by several physical parameters, including median aerodynamic diameter (MAD), size distribution, terminal velocity, dynamics and dynamics regime, partitioning, activation and coagulation [38]. Both the pressure and volume of an aerosol are affected by temperature [39]. Whereas the potential of therapeutic aerosols has been best investigated for decades in pulmonary medicine, there is relatively little knowledge about their use for intraperitoneal drug delivery.

We expected the development of therapeutic aerosols for intraperitoneal drug delivery to be easier than pulmonary applications for the following reasons: the geometry of the peritoneal cavity [40] is simpler than the geometry of the airways, which anatomically consists of a complex tree with successive divisions ending in numerous bronchioalveoles. Inertial impaction occurs chiefly in pulmonary medicine with larger particles whenever the transporting airstream is fast, changing direction, or turbulent (for example in the pharyngolarynx or at successive airways bifurcation). Inertial deposition is therefore influencing aerosol delivery by capturing a significant part of the therapeutic substance in the upper airways [41]. This problem is less important within the peritoneal cavity, where deposition mainly follows gravitational sedimentation. One of the most critical maneuvers during pulmonary administration is to coordinate the actuation of the aerosol with the patient’s inspiration. This problem does not exist during intraperitoneal administration, since no active patient collaboration is required.

### 4.3. Adding Hyperthermia

A cytotoxic effect of hyperthermia has been observed as early as the 19th century, when patients with malignant tumors experienced a regression of these tumors during high-febrile bacterial diseases [42]. Hyperthermia induces protein damage leading to apoptosis, repair by molecular chaperones, proteolysis or macroautophagy [43]. During the last two decades, therapeutic hyperthermia (between 41–43 °C) has been used for enhancing the cytotoxic effect of intraperitoneal chemotherapy [44] in combination with cytoreductive surgery (CRS). For this purpose, the abdomen is connected to an extracorporeal pump with a heat exchanger and chemotherapy is administered through a closed system of inflow and outflow catheters. CRS with HIPEC has developed to be a standard of care in selected patients with peritoneal surface malignancies [45]. However, there is no randomized trial validating the incremental clinical benefit of hyperthermia vs. normothermic intraperitoneal chemotherapy [46] and clinical results do not always meet the expectations [47].

## 5. Pressurized Intraperitoneal Aerosol Chemotherapy (PIPAC)

PIPAC is a minimally invasive approach relying on physical principles for improving intraperitoneal drug delivery, including: (1) optimizing the homogeneity of drug distribution by applying an aerosol rather than a liquid solution; (2) applying increased intraperitoneal hydrostatic pressure to counteract elevated intratumoral interstitial fluid pressure; (3) limiting blood outflow during drug application; (4) steering environmental parameters (temperature, pH, electrostatic charge etc.) in the peritoneal cavity for best tissue target effect. In addition, PIPAC allows repeated application and objective assessment of tumor response by comparing biopsies between chemotherapy cycles.

Staging laparoscopy has developed to be a standard of care in ovarian cancer [48] and in gastric cancer [49]. During such procedures, carbon dioxide (CO_2_) pneumoperitoneum is applied in order to create a working space. This working space allows for the safe placement of access ports through the abdominal wall [50], visualization of organs [48,51] and completion of complex interventions [52,53]. Twenty years ago, we developed a first-generation device suitable for minimally invasive surgery procedures that allowed microdroplets of a therapeutic substance to be distributed into the pneumoperitoneum (CO_2_), creating a "therapeutic pneumoperitoneum" [54].

Over the years, in addition to the aerosolization of a drug, we found it useful to add two further components, namely steering the operating environment (pressure, temperature, electrostatic charges, etc.) and objective assessment of tumor response (by comparing biopsies between repeated applications) [55,56]. All three components are now part of PIPAC [57]. Starting with the first preclinical experiments through technology development, first in-human use [58], Phase 1 [59,60,61] and Phase 2 [62,63,64,65,66,67] trials, PIPAC is currently being evaluated in randomized controlled trials [68,69,70,71,72] for palliative therapy of PM [73].

PIPAC (Figure 3) is applied through laparoscopic access using two balloon trocars in an operating room equipped with laminar air flow. In a first step, a normothermic capnoperitoneum is established with a pressure of 12 mmHg. A cytotoxic solution (about 10–20% of a normal systemic dose) is nebulized with a micropump into the abdominal cavity and maintained for 30 min. The aerosol is then removed through a closed suction system.

In contrast to inhalers commonly used in pulmonary medicine, no propellant gas is needed. During PIPAC, a liquid solution is aerosolized into the gaseous (CO_2_) environment using a specific nozzle (Capnopen®, Capnomed, Zimmern, Germany). Energy is provided by applying an upstream mechanical force gradient provided by an industry-standard angioinjector (e.g., Accutron HP®, MedTron, Saarbrücken, Germany).

Theoretical considerations suggest that the therapeutic CO_2_-pneumoperitoneum (“capnoperitoneum”) should be capable of carrying microdroplets of active substances to all exposed peritoneal surfaces. These considerations were confirmed by several preclinical experiments, showing that the active principle is distributed, reaching all exposed [28,74] and even partially hidden surfaces [75].

Figure 4 provides a graphical abstract of the differences between non-pressurized, liquid intraperitoneal chemotherapy vs. pressurized, aerosolized intraperitoneal chemotherapy.

As shown in the upper left panel, during liquid intraperitoneal chemotherapy, gravity forces applied to the tumor node depend on the height of the water column (h); the exposition of the tumor node depends on the filling volume. This reflects the situation of open HIPEC. In contrast, during PIPAC (upper right panel), tumor nodes are exposed to unidirectional gravitational and multidirectional hydrostatic forces caused by the pressurized environment. Although an aerosol gradient exists, all exposed peritoneal surfaces are reached by the drug. The net fluid transfer, as illustrated in the lower panels, results from the balance between gravitational and hydrostatic forces on the one hand and tumor node resistance resulting from increased interstitial fluid pressure on the other hand. As shown in the lower left panel, since resistance in normal tissue is lower than in tumor nodes, liquid chemotherapy will follow the path of least resistance and preferentially enter the normal tissue. In contrast (right lower panel), aerosolized chemotherapy will penetrate both the normal and the tumoral tissue, penetrating the normal deeper than the tumoral tissue.

## 6. Preclinical Evidence of PIPAC

Several studies have evaluated drug uptake into the target tissue after PIPAC, and some of them have compared PIPAC with other drug delivery techniques such as HIPEC and/or laparoscopic HIPEC. Therapeutic substances investigated include methylene blue [28,76], DNA [77], RNA [29,78], cisplatin [79,80,81,82], doxorubicin [75,83,84,85], oxaliplatin [82,86,87,88] and liposomal doxorubicin [89,90].

### 6.1. Effect of Hydrostatic Pressure

Increasing intraabdominal hydrostatic pressure during PIPAC significantly enhanced tumor cell toxicity of oxaliplatin in both wild-type and chemotherapy-resistant cells in vitro. A maximum cytotoxicity was observed at 15 mmHg. Pressures >15 mmHg did not show additional cytotoxic effect on cells [88]. Similarly, increasing PIPAC pressure over 15 mmHg did not increase the depth of doxorubicin tissue penetration in an ex vivo model [84].

### 6.2. Homogeneity of Distribution

Delivery of methylene blue into a large animal model unraveled a more homogeneous and more intensive vital staining of the abdominal cavity after aerosolization than after lavage [28]. Ex vivo experiments on surgical specimens of human peritoneal metastases evidenced a homogeneous distribution of small DNA fragments (Dbait) onto the peritoneum as compared to conventional peritoneal lavage [77]. In a rodent model, bioluminescence analysis showed a superior distribution of RNA complexes after intraperitoneal nebulization as compared to intraperitoneal injection of liquid: bowel exposition to the therapeutic substance was superior after PIPAC (median: 50%) vs. intraperitoneal injection (median: 10%) [29]. In a postmortem swine model, doxorubicin administered as PIPAC reached all areas within the peritoneum, with the highest depth of penetration being measured in the area located in front of the aerosolizer [85]. In two studies, the aerosol reached covered peritoneal areas [75,84]. In a living swine model, homogeneity of drug repartition (as defined by the comparison of the oxaliplatin concentration in the parietal vs. visceral peritoneum) was better after PIPAC (ratio: 11.5) than after HIPEC (ratio: 17.6) [30].

### 6.3. Tissue Concentration

All pharmacological data published so far show a superior therapeutic ratio (tissue concentration/dose applied) of PIPAC vs. systemic administration [29], of PIPAC vs. intraperitoneal liquid chemotherapy [29], of PIPAC vs. HIPEC [86] and of PIPAC vs. laparoscopic HIPEC [30]. In the swine model, tissue delivery of oxaliplatin was compared between laparoscopic HIPEC (with 12–15 mmHg pressure), PIPAC (20% of HIPEC dose applied) and electrostatic precipitation PIPAC (ePIPAC, same dose as PIPAC): overall concentrations in the peritoneum were not different among the three groups, documenting an improvement of the therapeutic index (target tissue dose/dose applied) of PIPAC vs. HIPEC by a factor of five. In the visceral peritoneum, the improvement of the therapeutic index reached a factor of 7.4 in favor of PIPAC (*p*  =  0.02).

### 6.4. Depth of Tissue Penetration

Most studies on depth of tissue penetration have been performed with doxorubicin, a large auto-fluorescent molecule with a molecular weight of 543 g/mol [91]. Doxorubicin only penetrates a few cell layers after HIPEC [92], corresponding to a depth of 10–30 µm. After PIPAC, depths of doxorubicin tissue penetration reached up to 469 µm in the ex vivo model [84], more than 300 µm in the small intestine of swine post-mortem [85] and up to 500 µm in human patients [58].

### 6.5. Local Effect vs. Systemic Uptake

Since the dose applied is reduced by an order of magnitude (as compared to a usual systemic dose) [93] and due to the properties of the peritoneum–plasma barrier, the plasma concentration of the chemotherapeutic agent remains low [57] and, accordingly, the organ toxicity [70]. In addition, pressure enhances the pharmacokinetic advantage of regional delivery by reducing blood outflow from the abdomen over the liver and the abdominal wall during the uptake phase. Thus, during the time of PIPAC exposition (currently around 30 min) at a pressure of 12–15 mmHg, the abdominal vascular compartment is expected to be partially isolated from the systemic compartment, contributing to better local uptake and less systemic toxicity.

This hemodynamic effect of pressure during laparoscopy has been documented in the preclinical model and in the human patient. Specifically, in a swine model of laparoscopic nephrectomy at a pressure of 12 mmHg, hepatic perfusion (both arterial and portal) was reduced by half: a decrease of portal flow from 974 mL/min to 547 mL/min and a decrease of hepatic artery flow from 278 mL/min to 133 mL/min was seen [94]. When splanchnic blood flow was measured at an increasing intraabdominal pressure from 0 mmHg to 15 mmHg in 18 patients undergoing routine laparoscopy, portal venous flow was decreased by 39% and microcapillary flow in the parietal peritoneum by 60% [95]. In the swine model, systemic oxaliplatin concentrations were significantly lower during PIPAC application time (in the presence of an intraabdominal pressure of 12–15 mmHg) than in the laparoscopic HIPEC group (*p*  <  0.05) [30]. In vitro, the rate of apoptotic and proliferative cells as well as the level of oxaliplatin penetration in tumor nodes was higher in PIPAC groups with less systemic passage through the peritoneum [86]. In vivo, in a mouse model of colorectal PM, systemic passage was lower in the PIPAC group [86].

## 7. Clinical Evidence on PIPAC in Chemoresistant PM

Preclinical and clinical evidence on PIPAC confirms that optimizing the physical parameters of intraperitoneal chemotherapy helps to circumvent initial or acquired resistance of PM to chemotherapy.

### 7.1. Local Efficacy

In pretreated patients, an objective clinical response of 62–88% was reported for patients with ovarian cancer (median survival of 11–14 months), 50–91% for gastric cancer (median survival of 8–15 months), 71–86% for colorectal cancer (median survival of 16 months), and 67–75% (median survival of 27 months) for peritoneal mesothelioma [96]. This clinical benefit is probably explained by local dose intensification: a high doxorubicin concentration in the target tissue was reported early in the PIPAC experience [58]. This was confirmed by a formal pharmacological study showing that doxorubicin and cisplatin were taken up in ascites and diseased peritoneum with an accumulation of doxorubicin but not cisplatin after repeated PIPACs [72].

### 7.2. Local Toxicity

Adverse events (Common Terminology Criteria for Adverse Events) greater than Grade 2 occurred after 12–15% of PIPAC procedures, which appears to be low as compared to catheter-based intraperitoneal chemotherapy. These adverse events commonly included bowel obstruction, bleeding, and abdominal pain. No chemical bowel perforations were reported in the literature. In contrast to catheter-based chemotherapy [97], no device-related complications were noted after PIPAC. However, severe peritoneal sclerosis was reported in two individual cases after repeated application of 92 mg/m^2^ oxaliplatin as PIPAC [98]. The importance of such long-term complications has to be weighed against a potential survival advantage in the salvage situation; the risk of peritoneal sclerosis is an argument for avoiding high-dose PIPAC in the prophylactic setting [64]. Notably, such long-term complications were not reported about repeated application of (low-dose) doxorubicin and cisplatin as PIPAC [98].

### 7.3. Systemic Toxicity

Plasma concentrations of doxorubicin after PIPAC peaked at 4.0–6.2 ng/ml after 30 min with a fast plasma clearance of 2.6–6.0 mL/min, corresponding to 1% of the Area Under the Curve (AUC) measured after systemic application of a usual dose [58]. A formal Phase 1 dose escalation study confirmed that serum peaks of doxorubicin and cisplatin after PIPAC sharply increase and peak after 30–45 min with a subsequent sharp decline, and with the drugs being totally cleared from the circulation after 120 min to 24 h, again depending on the dosage level [70].

Typical systemic side-effects of chemotherapy such as hematological, renal, cardiac, skin or hepatic toxicity or alopecia were not reported after PIPAC. Repeated PIPAC did not have a negative effect on quality of life [96,99].

In summary, the available clinical evidence demonstrated that PIPAC is feasible and safe. Data on objective response and quality of life are encouraging. Thus, PIPAC can be considered as a treatment option for chemotherapy-refractory, isolated PM of various origins [96,99].

## 8. Limitations of PIPAC

However, PIPAC technology is still in its infancy and there is a significant potential for improvement. Some issues are technical and related to the aerosolizing device. With current PIPAC technology, impaction forces generate a concentration gradient from the axis of the nebulizer to the periphery [100]. During PIPAC, the aerosol consists of a bimodal volume-weighted particle size distribution with a median aerodynamic diameter (MAD) of 25 µm. While the vast majority of droplets delivered during PIPAC have a diameter around 3 µm, over 97% of the volume of the aerosolized liquid is delivered as droplets with ≥3 µm in diameter. These larger droplets are primarily deposited by impaction and gravitational settling onto the peritoneal surface facing the aerosolizing device [100].

Thus, homogeneity of drug distribution after PIPAC is not ideal. Several studies ex vivo and in animal models have shown an increasing drug gradient from the top to the bottom of the target volume, as well as a decreasing gradient from the axis of the aerosolizing device towards the periphery [84,85,86,88,101]. The addition of electrostatic precipitation (so-called ePIPAC) reduced the influence of gravity and improved homogeneity of drug repartition. After ePIPAC, tissue drug concentration increased towards the electrode generating the electrostatic gradient [30].

It has been claimed that the drug distribution gradient observed after PIPAC is a disadvantage [84]. However, no clinical proof has been delivered for this claim. In fact, 70% of the peritoneal surface is visceral and the abdominal organs are located posteriorly when the patient is in the supine position, which is the case during PIPAC. Moreover, the limiting factor for reaching complete CRS is the serosa of the small bowel and this is the location where the relative pharmacological advantage of PIPAC over HIPEC has been documented to be maximal. This pharmacological advantage does not seem to be linked to more local complications: no chemical bowel perforations after PIPAC have been reported in the literature. Thus, we make the hypothesis that the concentration gradient observed after PIPAC might be beneficial in the clinical setting. Further clinical trials with advanced aerosolizing devices are needed to clarify this question.

Current PIPAC technology allows for aerosolization of solutions with higher viscosity, including polymers, carbohydrates, proteins, nucleic acids and lipophilic solutions. Moreover, the current technology has been shown to work in environments highly saturated with humidity, which was not possible with aerosolizers based on micro-perforated membranes [54]. Similarly, whereas endoscopic microcatheters are able to spray aqueous solutions [102], they cannot reliably aerosolize more advanced carbohydrate- or polymer-based formulations with higher viscosity.

Other issues are of an anatomical nature. Some closed spaces within the peritoneal cavity, in particular the lesser sac, are barely accessible to the therapeutic aerosol [40]. In patients in whom the lesser sac had not been opened during a previous surgery, isolated tumor progression was observed in this closed anatomic space when complete macroscopic remission was documented in the remaining abdomen [103]. Further obstacles to the diffusion of the aerosol are enteroenteral and enteroparietal adhesions, since most patients treated with PIPAC were in the salvage situation and had received previous surgery. In 10.5% of patients, such adhesions even prevented creation of an adequate working space, resulting in the abortion of the intended PIPAC procedure [96]. This is indeed an argument to perform PIPAC in patients with PM earlier in the course of disease and patients developing recurrence after CRS and HIPEC are not good candidates for the PIPAC procedure.

## 9. Electrostatic Precipitation Pressurized Intraperitoneal Aerosol Chemotherapy (ePIPAC)

An attractive path for further development of PIPAC is electrostatic precipitation of the therapeutic aerosol in order to improve the homogeneity of spatial distribution and depth of tissue penetration. In addition, ePIPAC has the potential to shorten the operating time needed for application and to reduce potential occupational health safety hazards [76]. Electrostatic precipitation devices have been certified for clearing the visual field from surgical smoke during laparoscopy [104] and the same technology can be used for precipitating therapeutic aerosols. We have first shown the in vivo feasibility of ePIPAC in the swine [76]. ePIPAC probably does not significantly increase tissue drug concentration so there are no additional concerns on local toxicity vs. PIPAC [30]. The first terminally ill patients treated with ePIPAC did not present with any significant adverse events, and radiological tumor responses were observed [105]. Clinical results of a first cohort of 135 PM patients treated with ePIPAC have been published: ePIPAC was well tolerated and safe. After three procedures and concomitant chemotherapy, response or stable disease was achieved in approximately half of cases [106]. Recently, ePIPAC with oxaliplatin was found to be safe in a prospective trial in a homogeneous patient cohort with colorectal PM, but the objective tumor response after 1 min exposition was below expectations [107].

## 10. Hyperthermic Pressurized Intraperitoneal Aerosol Chemotherapy (hPIPAC)

PIPAC allows for modification of the intraabdominal or intrapleural temperature by applying cooled or heated CO_2_. A dedicated device for generating hyperthermia was developed and tested in a large animal model. Using an extracorporeal heat exchange system, it was possible to apply cisplatin under moderate hyperthermia (38.8–40.2 °C) for 80–110 min; the animals tolerated the procedure well and organs were intact at autopsy [79]. To our knowledge, hPIPAC has not received regulatory approval and clinical application of hPIPAC has not been reported so far.

## 11. In Silico Modelling

Determining the key quality attributes for optimal intraperitoneal drug delivery would require a countless number of bench experiments. In silico simulation, i.e., using computational fluid dynamics models of drug delivery and interstitial fluid pressure, offer the possibility to model the influence of physical parameters on tissue drug delivery. By providing insights into the effect of modifying these parameters on peritoneal drug uptake, in silico modelling limits the need for in vivo experiments [108]. However, such simulations do not completely alleviate the need for confirmatory experimental measures. For example, in vivo imaging of vascular perfusion can visualize the uptake of therapeutic agents, as well as their spatio-temporal distribution within tumors [109]. Indirect confirmation can be obtained by matching in silico simulation results with high-resolution optical images of tumor tissue obtained in animal models [110]. Direct confirmation is best achieved by measurement of tissue drug concentration using established methods such as liquid chromatography (HPLC) or in situ mass spectrometry.

## 12. Conclusions

Theoretical considerations, as well as comprehensive preclinical and clinical data, suggest that optimizing the physical parameters of intraperitoneal drug delivery might help to circumvent initial or acquired resistance of PM to chemotherapy. Although incompletely understood, the reasons for PIPAC to overcome established chemoresistance are probably linked to local dose intensification. In the initial introduction phase, PIPAC has been used in patients who were quite ill and had already failed multiple treatment regimes, but it may not be limited to that group of patients in the future. Rapid diffusion of PIPAC in clinical practice worldwide supports its potential to become a game changer in the treatment of chemoresistant, isolated PM of various origins.

## Figures and Tables

**Figure 1 cancers-12-00034-f001:**
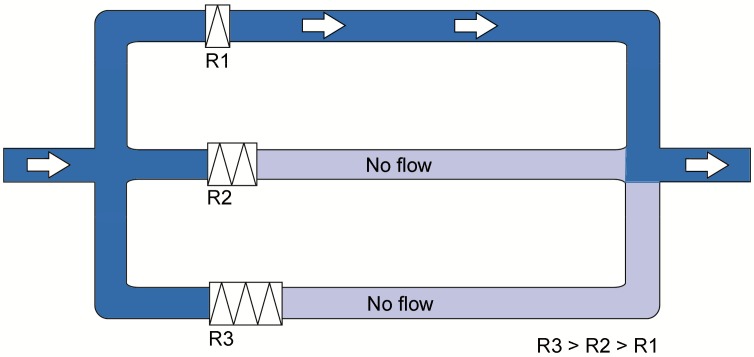
Influence of resistance on liquid flow in parallel circuits. A liquid always follows the path of least resistance. Three parallel pipes are exemplified. Assuming a non-limiting diameter of the pipes and where resistance (R)2 and R3 are higher than R1, all fluid will follow Circuit 1. This remains true even if the difference in resistance between Circuits 2 and 3 vs. Circuit 1 is minimal. The relationship between R2 and R3 plays no role.

**Figure 2 cancers-12-00034-f002:**
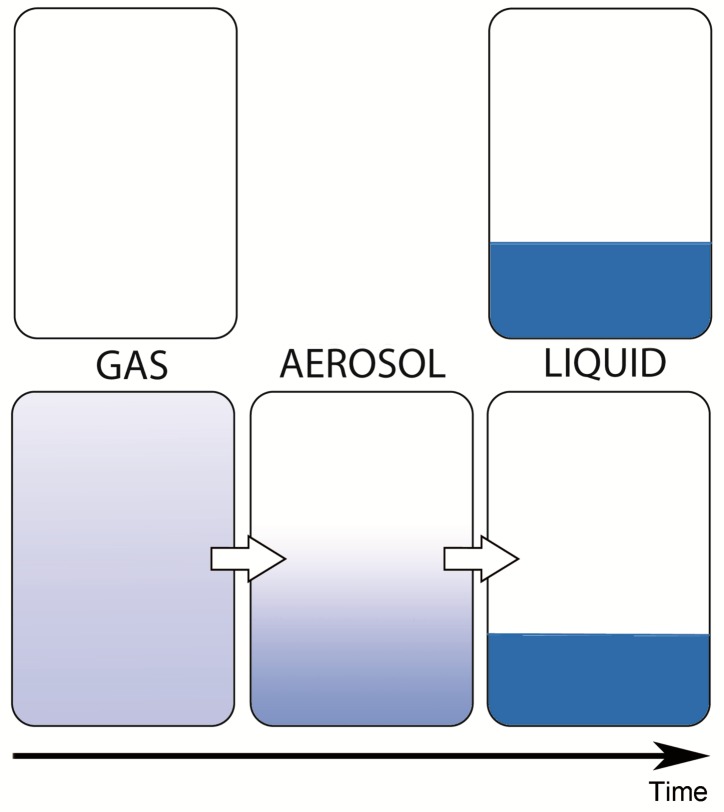
Behavior of a gas, a liquid and an aerosol within a closed volume. An aerosol is a suspension of liquid droplets within a gas. A gas expands homogeneously within a close volume, a liquid does not (assuming a constant temperature). An aerosol has an intermediate behavior. The smaller the droplets, the closer the behavior is to a gas; the larger the droplets, the closer the behavior is to a liquid. Due to gravitational force, every aerosol has a vertical concentration gradient and will sediment over time to the bottom of the container.

**Figure 3 cancers-12-00034-f003:**
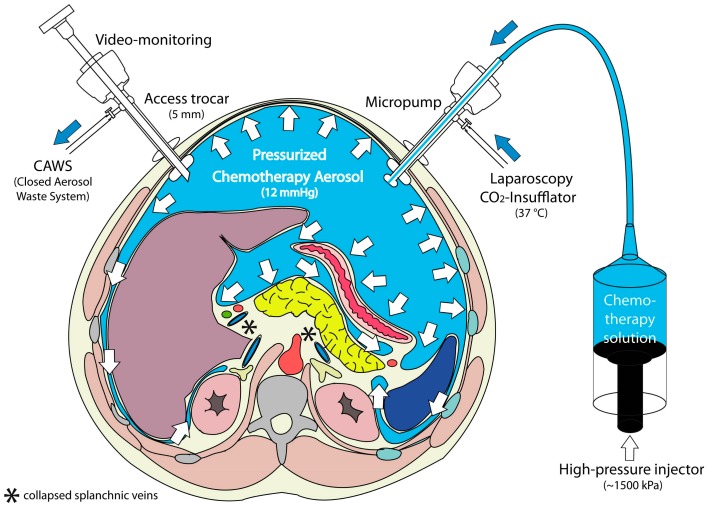
Principle of Pressurized Intraperitoneal Aerosol Chemotherapy (PIPAC). During a staging laparoscopy, an aerosol cytostatic agent is applied in the abdominal space using a nebulizer. The application as an aerosol allows the relatively even distribution of the substance. Increased pressure (12 mmHg) ensures deeper penetration into the tissue. Reproduced with permission from [58].

**Figure 4 cancers-12-00034-f004:**
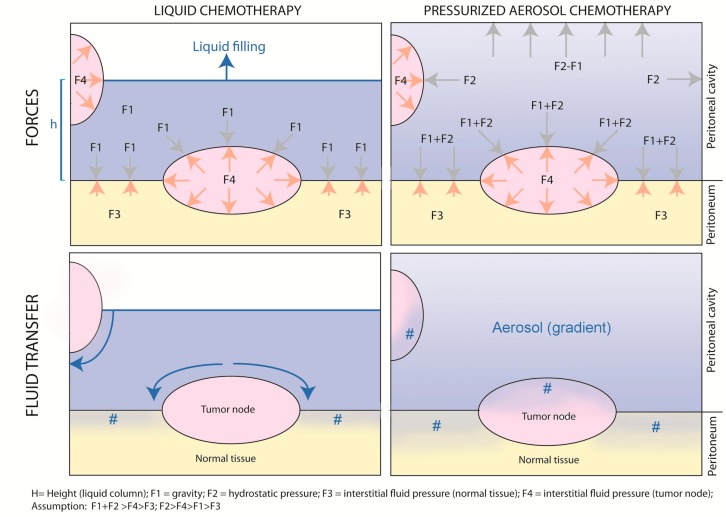
“Intraperitoneal chemotherapy with normobaric liquids vs. pressurized aerosols. Graphical abstract of the physical differences between non-pressurized, liquid intraperitoneal chemotherapy vs. pressurized, aerosolized intraperitoneal chemotherapy.” #: drug penetration into the tissue.

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
