# Peer review of "Overcoming Drug Resistance by Taking Advantage of Physical Principles: Pressurized Intraperitoneal Aerosol Chemotherapy (PIPAC)"

_cancers, 2019, doi:10.3390/cancers12010034_

Round 1
Reviewer 1 Report
Overall this is a review highlighting a new method of intraperitoneal chemotherapy. While I do not have extensive familiarity with this subfield of chemotherapy treatment, the structure of this review seems to significant gaps. Furthermore, the tables and data presentation should be considered such that they are used to communicate information to the reader most effectively. For example: Table 1 is essentially a list. This should be integrated with additional information or converted into the text as bullet points or section headings.
The structure of this review is not focused and could be greatly improved by clearly addressing the main focus sooner. Additionally there are many areas that are mentioned/included, but are not given the appropriate level of detail. Many are not conceptually highly tied to the title. For example, if this review is focused on chemotherapy delivery methods, the introduction should focus more on that than on diagnoses.
Reviewer 2 Report
Manuscript “overcoming drug resistance by taking advantage of physical principles: Pressurized Intraperitoneal Aerosol Chemotherapy (PIPAC).
The authors report a review of physical principles of intraperitoneal chemotherapy on peritoneal metastasis.
This manuscript is very interesting. In surgical oncology, these reviews of physiologic principle of therapeutic are unfrequent, despite their important role to built futur research and evidence base medicine.
This manuscript is clear and well written.
Comments
The discussion of authors about better distribution of aerosol over liquid solution is incomplete. Peritoneal adherences due to previous surgery and/or metastasis infiltration are frequent in patients with peritoneal carcinomatosis. In this setting, the geometry of the peritoneal cavity is not so simple as described in p 9 line 258 or similar to healthy animal model (ref 42). Some studies showed that the distribution of aerosol of PIPAC is not so homogeneous in the abdominal cavity. Clinical advantages of aerosol distribution over liquid solutions is not so clear and authors should specify these points. Fig 4 should be clarify with additional comments. The authors consider that no hydrostatic pressure exist with liquid chemotherapy. During HIPEC, more than 3l of liquid solution is frequently used. I don’t understand that this important volume has not hydrostatic pressure. The authors should clarify this point.Author Response
Please see the attachment

Reviewer 3 Report
The paper is a good combination of present and future treatment aspects in terms of PM.
I feel there could be more explanation given in the discussion section about how this therapy could be proven beneficial.
Figure one and two can be made more descriptive.
Rest of the paper I feel is very informative and well written.
Round 2
Reviewer 1 Report
The manuscript is significantly improved. My only suggestion would be that the abstract was unchanged despite the significant changes to the manuscript. I think the abstract could be improved to better reflect the improved manuscript.
Author Response
The abstract has been updated according with the reviewer's recommendation.